# A Retrospective Investigation on Age and Gender Differences of Injuries in DanceSport

**DOI:** 10.3390/ijerph16214164

**Published:** 2019-10-29

**Authors:** Jerneja Premelč, Goran Vučković, Nic James, Lygeri Dimitriou

**Affiliations:** 1Faculty of Sport, University of Ljubljana, 1000 Ljubljana, Slovenia; goran.vuckovic@fsp.uni-lj.si; 2Faculty of Science and Technology, Middlesex University, The Burroughs, Hendon, London NW4 4BT, UK; n.james@mdx.ac.uk (N.J.); l.dimitriou@mdx.ac.uk (L.D.)

**Keywords:** injuries, dance, ballroom, incidence, age

## Abstract

In spite of the extensive research on incidence site and type of injury in ballet and modern dancers, limited studies on injury in DanceSport have been reported. Therefore, this study determined retrospectively (within last 12 months) incidence, severity, site and type of injury, between gender and age-class in DanceSport. Participants were 97 international sport-dancers (female, 41; male, 56). Sixty-six (69%) dancers reported 96 injuries (1.00 (range = 4)) injuries per dancer) and an injury incidence of 1.7 (range = 14) per 1000 h. Females revealed significantly higher median injury incidence (females, 2.6 (range = 14); males, 1.9 (range = 9), *p* < 0.05) than males. A total of 61.5% of all injuries recorded were traumatic with a significant gender difference (Wald chi-square = 11.616, df = 1, *p* < 0.01). Injury severity was 3 (range 240) days with an interaction effect between gender and age-class (Wald chi-square = 251.374, df = 3, *p* < 0.001). Meanwhile, 72.3% of the dancers reported not including sport specific exercises besides dancing. These findings show gender and age-class differences in injury incidence, type and severity. Therefore, to reduce the likelihood of injuries, the implementation of supplemental DanceSport specific exercises that also considers the gender and age-class anatomical, functional, and choreographic demand differences in the training program should be recognized.

## 1. Introduction

DanceSport consists of three different disciplines: Standard dances (waltz, tango, Viennese waltz, slow foxtrot and quickstep), Latin-American dances (samba, cha-cha-cha, rumba, paso doble and jive) and ten dances (five standard and five Latin-American dances) [1]. DanceSport demands high aerobic and anaerobic energy [2,3], static and dynamic strength, core stability and balance [4,5]. During competition couples dance minimum from five to 20 dances, each dance lasts approximately 1.5 to 2 min and all five dances are performed consecutively with 1 min break between each dance. Blanksby and Reidy [6] found sport-dancers to dance higher than 80% of their maximal oxygen uptake and reported mean heart rates of 170 bpm (males) and 173 bpm (females) during standard dances and 168 bpm (males) and 177 bpm (females) during Latin-American dances. They also reported very similar mean gross oxygen uptake values between males (42.8 ± 5.7 mL.kg-lmin-1) and females (42.8 ± 6.9 mL.kg-1min-l) during standard and Latin American dances, respectively. Blood lactate values during competition in sport-dancers (9.89 ± 3.39 mmol/l) are comparable to that recorded during ballet, field and racquet sports [7,8] suggesting high intensity anaerobic levels. These studies suggest that for sport-dancers to avoid discrepancies between the physiological demands of competition and dance training, they need to train also at high exercise intensities that meet the physiological demands imposed during competition. To our knowledge, there are no studies yet, that describe the training regimes besides sport-dancing and investigate the competition demands versus DanceSport specific training. Angioi’s et al. [9] showed a discrepancy between performance and fitness levels and emphasized that this should be considered in the strength and conditioning of dancers to meet the choreographic demands. Such discrepancies have been suggested to be associated with increased likelihood of injury [4,10,11]. However, there is little consistency between studies in dancers investigating injury. Discrepancies include methodological designs such as retrospective versus prospective injury data collection, level of dancers, and type of dance [12,13,14,15]. Retrospective analysis (1991–2007) from the National Electronic Injury Surveillance System in the US emergency department showed that 113.084 injured dancers (3–19 years old) were mostly (55%) from ballet, modern, jazz and tap dance [16]. The most commonly reported dance injuries are sprains or strains of the lower limb [12,14,17]. Analogous to other sports most dance injuries are overuse rather than traumatic [5].

To our knowledge, the only relevant studies in DanceSport have used self-reports. Riding McCabe et al. [18] studied 99 sport-dancers, they showed that the most reported injured body sites were the knee (17%) and lower back (14%). Pellicciari et al. [19] studied 153 sport-dancers; they found 47.7% to report being injured in the last 12 months and to have sustained a total of 102 injuries. The most common injured sites reported were the lower limb (34.3%), ankle (23.5%) and knee (15.7%). Kuisis et al. [20] found an incidence of 0.99 injuries per 1000 h for all standard dance age groups over a period of 12 months. The most common pain complaints during dance training, in 13-year-old beginner standard dancers, have been reported to occur at the upper back (41%), the ankle and the foot (38%) [21]. In DanceSport similar results have been reported in junior and senior male dancers; upper back (42.7%); toe (44.9%); upper back (36.7%) and lower back (30.6%) in females [22]. 

Studies have investigated injuries for different age groups [23], such as identifying the impact of biological body changes and training load and intensity on injury occurrence [24,25]. Similarly, Landerson et al. [26] reported a positive relationship between age and injury incidence in adolescent ballet dancers whilst Yung et al. [27] found an increased injury risk in adolescents as a result of higher training loads and increased training intensity. 

In spite of the extensive research on incidence, site and type of injury in ballet and modern dancers, limited studies investigating injury incidence and differences between gender and age in DanceSport have been reported. Therefore, the primary aim of this study was to determine retrospectively (within last 12 months) the site, type, incidence and severity of injuries reported between gender and age-class in DanceSport. The secondary aim was to determine the type of activity and perceived cause of injuries sustained. 

## 2. Materials and Methods 

### 2.1. Ethics 

All volunteers gave their informed consent for inclusion before they participated in the study. This study was approved by the Ethics Committee of the Faculty of Sport at the University of Ljubljana. The study’s objectives and methods were explained to each participant, before a written, informed consent was obtained.

### 2.2. Participants

One hundred and one dancers from 21 different countries, out of 212 participating in an international DanceSport competition volunteered to participate in this study. The inclusion criteria were: Dancers had to be at the international level age 12 to 53 years, which were divided into four World DanceSport Federation’s (WDSF) [1] age-classes; junior (12–15 years), youth (16–19 years), adult (20–29 years) and senior (30–35 years), leaving 97 participants (41 female, 56 male) in the study. 

### 2.3. Questionnaires

Questionnaire details were verbally explained to each participant in conjunction with written instructions. Whilst participants were completing these investigators were present to clarify any issues/questions and tried to ensure that the answers were as accurate as possible. 

A simple self-administered questionnaire, internally designed, assessed the dancers demographic data including current body mass (kg) and stature (m); date of birth; gender; nationality and dance history; average weekly training volume (hours); total number of competitions and number of days off training during holidays in the past 12 months. 

Body mass index (BMI) was calculated as kilograms per square meter (kg.m^-2^). Adult dancers (age ≥ 20) were classified as underweight (<18.5 kg/m^2^) or normal (>18.5 kg/m^2^). The World Health Organization’s [28] classification was used to classify children and adolescents (age ≤ 19) as underweight (<3rd percentile) or normal (>3rd percentile). 

Training volume (hours) in the last 12 months were determined by multiplying the average dancing hours per week with dancing weeks per year. 

### 2.4. Injury History Questionnaire

The injury history questionnaire assessed number of injuries, type and site of injury, days off training due to injuries, type of activity injury sustained (training or competition), perceived cause of injury and history of treatment received in the last 12 months. Dancers were also asked whether they had ever been provided with free medical examination and if their dance club had a club doctor or therapist. The questionnaires that were used to assess injury [29,30] were combined and amended according to this study’s objectives.

Injuries were reported using a time-loss definition of injury, as modified from Brooks et al. [31]. An injury was only recorded when a musculoskeletal complaint sustained by a dancer during training or competition prevented the dancer from taking full part in all training or competitions for more than one day following the day of injury, irrespective of whether training or competitions were scheduled [29,30,31]. Injuries were reported as the total number of injuries and the mean number of injuries per dancer during the last 12 months. 

Injury incidence was reported as injuries per 1000 h (Allen et al.) [15]. Injuries sustained per site of injury were reported as absolute number, as relative number to all injuries sustained and as absolute number of injuries per 1000 h in the last 12 months. Severity of injury was classified according to the number of days off training and/or competition because of injuries [15,29] in the last 12 months. This definition was modified from studies by Brooks et al. [31], Fuller et al. [29], and Pluim et al. [32]. Injury severity was also grouped according to Allen et al. [15] who characterize injuries as transient (return within seven days) and severe (return after 84 days). The absence was reported as relative number (days off training and/or competition due to injuries/absolute dancing hours) [15]. The type of injury was characterized as traumatic or overuse according to the Fuller et al. [29] definition. Traumatic is defined as an injury which resulted from single specific identifiable event, and overuse as an injury caused by repeated micro trauma without a single identifiable cause [29]. Absolute number of dancing years was also recorded. 

### 2.5. Data Analysis 

Statistical analyses were undertaken using SPSS V.21.0 software (SPSS Inc., Chicago, IL, USA). Values of anthropometric, injury incidence, severity, absence, dancing history and training volume were reported as median (range), since the data were not normally distributed (Kolmogorov-Smirnov test, SPSS V.21.0 software ((SPSS Inc., Chicago, IL, USA)). Site and type of injury, injury sustained according to the type of activity, perceived cause and history of treatment received were reported as percentage. 

A two-way full factorial generalized Poisson loglinear model was used to assess differences and significant interaction effects between gender (males versus females) and age-class (junior, youth, adult and senior) for anthropometric details, injury incidence, severity, absence, type of injury, site, dancing history and training volume. This model was used because the data were frequency counts with skewed distributions. The test statistics for this model is the Wald chi-square with a level of significance set at *p* < 0.05.

## 3. Results

Median age, height, weight and BMI of dancers was 20 (range = 41) years, 171 (range = 38) cm, 60 (range = 45) kg, and 20.5 (range = 12.92) kg/m^2^, respectively. Although a significant gender difference was found for height (Wald chi-square = 8.511, df = 1, *p* = 0.004) no significant differences in age-class (Wald chi-square = 5.600, d34f = 3, *p* = 0.133) or interaction effect between gender and age-class (Wald chi-square = 1.280, df = 3, *p* = 0.734) were reported. A significant gender (Wald chi-square = 33.744, df = 1, *p* < 0.001) and age-class difference (Wald chi-square = 28.878, df = 3, *p* < 0.001) was found for weight with no interaction effect between gender and age-class (Wald chi-square = 4.462, df = 3, *p* = 0.216). Only seven females and three males were classified as underweight with 87/97 (89.7%) of the dancers having normal BMI (Table 1).

A total of 68.75% of dancers reported injuries in the last 12 months. A median of 1.0 (range = 4) injuries per dancer, 1.7 (range = 14) incidence of injuries and a total of 96 injuries were recorded. A significant gender difference was found in number of injuries per dancer (Wald chi-square = 6.103, df = 1, *p* < 0.05) and incidence of injuries (Wald chi-square = 18.120, df = 1, *p* < 0.001). Females reported a same median and higher range of number of injuries per dancer (female, 1 (range = 4); male, 1 (range = 3) and a higher incidence of injuries per 1000 h (female, 2.6 (range = 14); male, 1.2 (range = 9)) than the males. No significant difference in age-class (Wald chi-square = 7.171, df = 3, *p* = 0.067) and no interaction effect between gender and age-class was found in incidence of injuries (Wald chi-square = 7.277, df = 3, *p* = 0.064, Table 2).

Median injury severity (number of days off training and/or competition because of injuries) and absence (days off training and/or competition due to injuries/1000 h) was 3 (range = 240) days and 5.2 (range = 729) days/1000 h, respectively. However, 42 (43.3%) dancers reported less than 7 days off training whereas six had more than 84 days off. A significant interaction effect between gender and age-class was found in severity of injuries (Wald chi-square = 251.374, df = 3, *p* < 0.001) and absence (Wald chi-square = 778.69, df = 3, *p* < 0.001), (Table 2).

Median dancing hours was 504 (range = 864) in the last 12 months. The dancing season included 48 weeks of training (10.5 (range = 18) hours per week) and several competitions (2 (range = 4) competitions per month). The median years of dancing was 10 (range = 21) although a significant age-class difference was found (Wald chi-square = 67.724, df = 3, *p* < 0.001) but no significant gender or gender and age-class interaction was observed (Table 2).

### 3.1. Traumatic/Overuse Injuries

Most injuries were classified as traumatic (61.5% overall) with a significant gender difference (Wald chi-square = 11.616, df = 1, *p* < 0.01) such that females sustained more traumatic injuries (74.6%) compared to males (46.7%, Figure 1). No significant age-class and gender differences were found for overuse injuries.

### 3.2. Type and Site of Injuries

The most common injuries recorded were spasm/strain/tear at neck muscles (22%), muscle spasm/strain/tear at lower back (18%) and joint/ligament derangement at knee (16%), see Figure 2. Significant differences were observed between genders in lower back muscles incidence (Wald chi-square = 8.797, df = 1, *p* < 0.05) and between age-class in the knee (Wald chi-square = 8.565, df = 3, *p* < 0.05).

### 3.3. Injury Sustained According to the Type of Activity, Perceived Causes and Treatment of Injuries

Dancers reported that most of their injuries occurred during training (73.6%) or in competition (26.4%). The highest perceived causes of injury were overtraining (25%) and insufficient warm up (17%), (Figure 3). The average duration of their warm-up was 18.9 ± 10.1 min for training and 29.8 ± 18.2 min during competitions.

Most dancers (72.3%) did not include any special physical conditioning besides dancing. Only 27.7% of the remaining dancers reported that they used jogging and pilates to improve their overall fitness. Only 19.6% of all dancers (injured and not injured) received free medical examination, which was part of their annual prevention treatment, in the last 12 months. Only 15.2% of the sport-dancers had their own doctor/therapist in the dance club and 96% needed to pay for their own treatment in case of an injury.

## 4. Discussion

This study examined retrospectively (within last 12 months) the type and site of injury, incidence, severity, and injury sustained according to the type of activity and perceived cause of injuries between gender and age-class in DanceSport.

The female dancers had significantly higher incidence (10.8% higher) and number of injuries per dancer (6% higher) compared to males. According to Sedgwick [33], differences greater than 5% have a clinical interest. Similar to our study, Kuisis et al. [20] reported a higher injury incidence in female sport-dancers (1.45 injuries per 1000 h) compared to males (0.49 injuries per 1000 h) in standard dance. Female sport-dancers have more extreme leaning back hold, execute a higher number of multiple rotations, and wear higher heeled shoes than male sport-dancers suggesting that these differences might contribute to an increased likelihood of injury incidence as seen here and in Kuisis et al.’s [20] findings. Furthermore, the anatomical, biomechanical and hormonal gender differences might have contributed to these results [34].

In youth category, the male dancers reported higher number and injury incidence at the lower back and neck compared to the other age-class groups. These differences might be a result of growth-induced alterations in the vertebrae and moment of inertia, mass and length of the extremities possibly leading to readjustment/relocation of the body’s center of gravity [24,25]. The mean height and mass of the youth male dancers was greater by 11 cm and 10 kg respectively than junior category which suggests growth. During growth or growth spurt, the stress imposed on the myotendinous and osteotendinous junctions, ligaments, and growth cartilage is increased and flexibility might also decrease as a consequence [25]. Furthermore, the increases in strength required to support these changes to enable, for example, a dancer to generate similar limb speeds as before the growth spurt might be insufficient, leading to growth and strength imbalances and an increased likelihood of adolescent injuries especially during training and competition [23,24,25]. DanceSport choreography demands vigorous repetitive hyperextension of the lumbar spine; this has been associated with low back pain [25]. Furthermore, in youth category the choreography becomes more complex and physically challenging as the frequency and speed of turns and hip movements is increased compared to junior category. This can be partly supported by previous research that showed significant differences in velocity between lower and higher rank dancers [35]. Apparently, no significant age-class or gender differences were observed for absolute dancing hours and percentage of dancers performing strength training to meet these increased physical demands, growth and strength imbalances. Additionally, the changes described above in synergism with impulsiveness and recklessness typically seen in teenagers, might increase the likelihood of injury [23]. Thus, the authors of this study recommend the implementation of supplemental exercises in the training program of sport-dancers and particularly of adolescents that suffer from higher incidence and severity of injuries, to maximize strength and possibly reduce the likelihood of injury associated with an increased physically demanding choreography, growth, and strength imbalances. 

### 4.1. Traumatic/Overuse Injuries

In this study the proportion of traumatic injuries was greater than overuse ones which contrasts with Kuisis et al. [20] who found similar numbers of traumatic and overuse injuries. DanceSport is characterized by explosive, fast and repetitive movements, sudden change of direction, turns, hopping, and kicking known to contribute to traumatic injuries [36]. Repeated traumatic injury, dancing in pain, overtraining, and insufficient recovery might lead to overuse injuries [37,38,39]. 

A significant gender difference was observed for traumatic injuries with females reporting more traumatic than overuse. This is surprising as previous research has consistently shown the opposite trend [37,38,39]. Female sport-dancers perform more extreme and quick back bending and head turns, and their choreography includes more movements from dancing to quick stop positions, with high lifted leg or back hyperextension compared to males. These might explain the higher number of traumatic rather than overuse injuries found in females. Within multidirectional movements, sport-dancers must maintain a degree of stability and balance when transitioning from a dynamic to static and back to a dynamic state. Dynamic stability could help a dancer to maintain a greater stable center of gravity during these dance-specific movements as seen previously in other sports [40,41]. It is of concern that only 27.7% of sport-dancers reported the use of other types of training to improve their overall fitness and the majority of injuries were sustained during dance training (73.6%). Therefore, it is important that DanceSport coaches ensure that specific exercises that develop technique, posture, hold, balance, coordination, static and dynamic strength [42,43], multidirectional speed, and dynamic stability are included in a sport-dancer’s conditioning program which could minimize the likelihood of injury [4,9]. The longitudinal effect of gender and age-class specific supplementary training should be investigated to determine if this could counter some injury factors. Furthermore, the determination of the onset of peak height velocity as a reference point for the design and implementation of supplement exercise and individualized training programs should be investigated. Previous research has shown that various physical performance attributes are related to biological maturation during male adolescence [44].

### 4.2. Body Sites of Injury

The most frequently injured body sites for both male and female sport-dancers were the neck muscle, lower back muscle spasm/strain/tear, and knee joint/ligament derangement which is similar to Kuisis et al.’s [20] findings (23% of all injuries in standard dance were located in the neck and back). When large compressive forces are applied on the neck especially during hyperextension the spine is forced out of its natural alignment and the likelihood of neck injury is significantly increased [7]. 

Riding McCabe et al. [18] and Pellicciari et al. [19] also reported most common injuries in sport-dancers were lower limbs and lower back. Koutedakis et al. [8] suggested that repetitive overstress on the anatomical structures can result in hypermobility and poor alignment of the knee joint. Similarly, the wearing of high heel shoes by female dancers exacerbates weight transfer forces [45]. which can subsequently increase foot, knee and back injuries [46]. 

The findings of this research could help dancers and their coaches to appreciate that sport-dancers are frequently exposed to injuries. They should be aware of dancers’ physical and anatomical limitations that exist between age-classes and gender. Special education for dance teachers on gender and age-related injuries should be considered. Injury prevention programs could include proper planned dance activity for different age-classes and gender to minimize the likelihood of injury and include supplementary physical preparation and specialized health care services. 

### 4.3. Limitations

Self-reported questionnaires based on retrospective reflections on injuries have been critiqued in terms of credibility of responses. Since injury questionnaires were completed by participant recall, this may call their accuracy into question. However, high levels of self-awareness seen in athletes [47] and the fact that many kept a training diary are likely to support accuracy. However, most published retrospective studies have examined the 12 months period that was also used in this study. We also acknowledge the lack of clinical injury records for our participants as well as data on alcohol and dietary intake, the latter of which has been associated with increased risk of injuries such as stress fractures. Nonetheless, the implications of these findings should not be underestimated.

Sample size and grouping dancers into four age-classes meant that some results were based on relatively low numbers. We used this classification because these age classes are based on the official age dance classification and each age-class is engaged in different training exercises and routines, which might be also associated with different types and body site of injuries and injury incidence.

## 5. Conclusions

In this study, significant differences were observed by gender and age-class for number of injuries, injury incidence and severity, type, and site of injuries. Undoubtedly there are gender and age-class anatomical, functional and choreographic demand differences with the majority of dancers reporting no engagement in sport-specific exercises that address these differences. It is therefore suggested that specific training in DanceSport should optimize the overall fitness of the sport-dancer and hence reduce the incidence and severity of injury. Further studies should focus on the impact of choreography demands, for different age-classes and gender, on injury incidence and severity.

## Figures and Tables

**Figure 1 ijerph-16-04164-f001:**
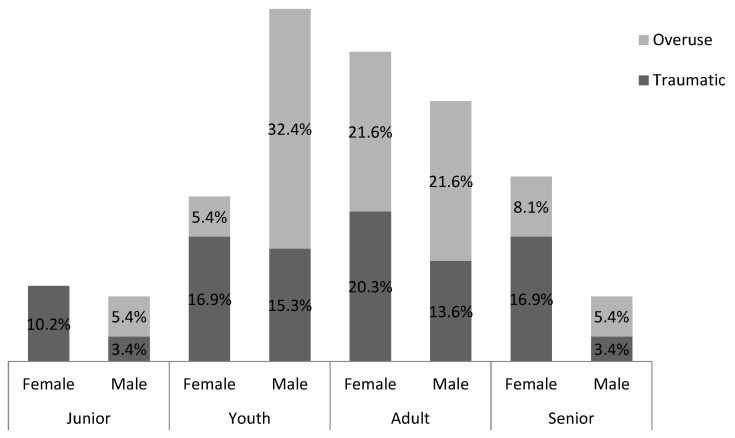
Percentage of traumatic and overuse injuries by gender and age-class.

**Figure 2 ijerph-16-04164-f002:**
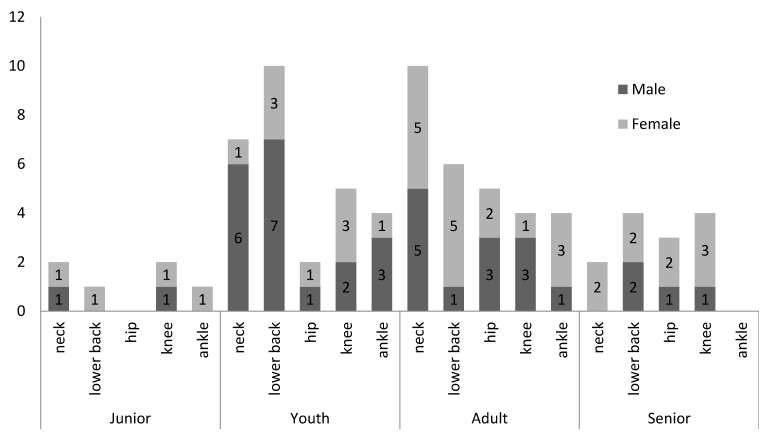
Most common injured body location by gender and age-class.

**Figure 3 ijerph-16-04164-f003:**
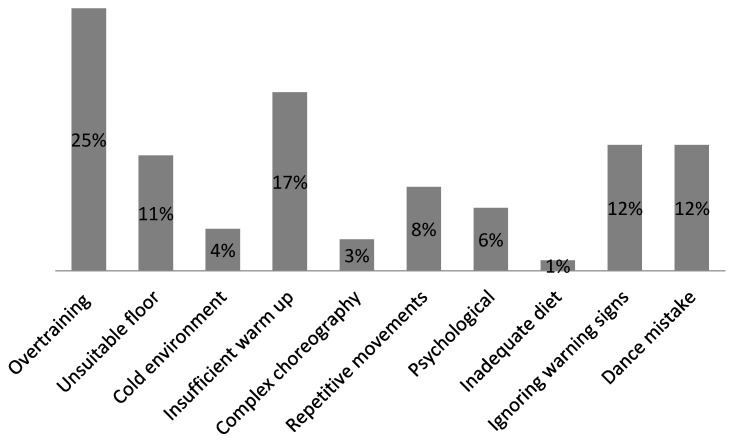
Percentage of perceived causes of injuries in sport-dance.

**Table 1 ijerph-16-04164-t001:** Descriptive characteristics of participants by different gender and age-class.

Age-Class	Gender	Junior	Youth	Adult	Senior
Number of	F	6	10	16	9
dancers	M	10	29	35	17
Age (yr)	F	14 (3)	19 (3)	22 (9)	37 (18)
	M	13 (2)	18 (3)	23 (9)	46 (21)
Mass (kg)	F	48 (19)	52 (13)	55 (31)	53 (25)
	M	50 (27)	60 (26)	69 (44)	73 (19)
Height (cm)	F	165 (14)	165 (19)	169 (32)	167 (21)
	M	161 (29)	176 (22)	177 (18)	178 (7)
BMI	F	17.6 (7.9)	19.1 (4.9)	19.4 (7)	20.3 (9.4)
	M	19.3 (6.8)	20 (5.9)	21.7 (12.2)	22.9 (5.3)

Values are median (range); *p* < 0.001.

**Table 2 ijerph-16-04164-t002:** Incidence and severity of injuries in the last 12 months between gender and age-class.

Age-Class	Gender	Junior	Youth	Adult	Senior
Number of injuries		10	33	36	17
Injuries per dancer^*^	F	1.0 (2)	1 (3)	1 (3)	1 (4)^**^
	M	0.0 (1)	1 (3)	1 (3)	0.5 (1)^**^
Incidence^†^	F	1.6 (5.2)	1.9 (13.9)	3.0 (9.3)	2.6 (13.9) ^**^
	M	0.0 (4.6)	2.1 (9.3)	1.0 (6.9)	0.7 (3.5) ^**^
Severity^‡^	F	3 (30)	3 (25)	3 (180)	10 (210)
	M	0 (5)	3 (210)	2 (60)	3 (240)
Absence^§^	F	4,7 (78)	6.1 (69)	9.1 (625) ^**^	19.8 (365)
	M	.0 (19)	6.9 (729)	3.0 (208) ^**^	2.6 (333)
Years of Dancing^‖^	F	5.5 (4)	8 (12)	12.5 (15)	9 (16)
	M	4 (2)	8 (10)	13 (21)	10 (18)
Dancing hours^¶^	F	13.3 (17)	10.5 (17)	10 (17)	8 (18)
	M	10.8 (13)	10 (17)	12 (15)	8 (9)

* Median number of injuries per dancer (*p* < 0.05); ^†^ median number of injuries per 1000 h of dancing; ^‡^ median number of days off training due to injuries per dancer; ^§^ median number of days off training per 1000 h of dancing; ^‖^ median years of dancing; ^¶^ median hours of dancing per week; ** significant differences. Values are median (range).

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
