# Peer review of "A Retrospective Investigation on Age and Gender Differences of Injuries in DanceSport"

_ijerph, 2019, doi:10.3390/ijerph16214164_

Round 1

Reviewer 1 Report

Check the paper for English. Although it's really pretty good, there are a few instances, where the phrasing is a little odd.

I really like this paper. It is chock full of information about a popular dance sport that has probably ignored its injury rate. 

Grouping the younger and older adults together might be somewhat problematic, for overall data figures. However . the authors do a nice job of looking at the younger and older groups separately in terms of understanding their injuries better. 

In the Discussion, please be careful not to restate the results. Very Briefly mention a result and then go on to discuss and relate to existing literature. 

The paper very nicely fills a huge gap. 

Author Response

Response to Reviewer 1 Comments

Dear reviewer,

Thank you very much for your feedback and suggestions. We try to improve our paper as much as we could and hopefully, our changes satisfied your expectation.

Grouping the younger and older adults together might be somewhat problematic, for overall data figures. However. the authors do a nice job of looking at the younger and older groups separately in terms of understanding their injuries better. 

We used official sportdance age classification. This is because these age classes are based on the age dance classification and it will make more sense to the dance community and the results will have more applicability. Furthermore, each age group is engaged in different training exercises and routines which might also be associated with different types of injuries, different body sites and therefore different incidence of injuries.

In the Discussion, please be careful not to restate the results. Very Briefly mention a result and then go on to discuss and relate to existing literature. 

The discussion, limitation and conclusion were rewritten. Thank you very much for all your suggestions that help us to improve our paper. We hope that we satisfied your expectations.

Kind Regards

Reviewer 2 Report

Thank you for your invitation to review this paper, which investigates an understudied genre of dance. Please find below some comments and recommendations for review: 

There is contention in the field regarding the classification of dance as only a performing art and not a competitive sport. Therefore I suggest deleting lines 29-30, and focusing only on explaining the role competitions play in DanceSport, rather than comparing this with ballet and modern dance. Overall the introduction is lengthy and at times the relevance to your specific research questions are unclear. Suggest to review and tighten this whole section. References need to be updated, as the most recent reference is from 2014. Some Dance Sport injury studies have been published since this time that are of relevance, including: Pellicciari et al 2016 “Injuries Among Italian DanceSport Athletes: A Questionnaire Survey”, and Riding McCabe et al 2014 “Fit to dance survey: a comparison with dancesport injuries” and Riding McCabe et al 2013 “A Bibliographic Review of Medicine and Science Research in DanceSport”

Methods:

Use of reference 26 on line 97: there is likely a more relevant reference to support the questionnaire used in the study (Fit to Dance perhaps?) Dance training volume information is duplicated in both the demographic data and the injury history sections, only needs to be in one. Please provide more information on how ‘hours danced in the last 12 months” was determined (for example was it the average hours spent dancing per week X weeks spent dancing per year?) Lines 122-25: presentation of injuries as a rate per 1000 dance hours provides standardisation of results across injury epidemiology studies. I disagree that it cannot (or should not) be applied to participants who train less than 1000 hours per year. Suggest deleting this section Severity of injury: please provide more details as to how many days off dance would be required for each severity ‘group’? Or were these data collected and analysed as a continuous variable? If so I suggest rewording this sentence, as data were not ‘grouped’.

Results:

Wording of the results section could be refined down quite a bit, much of this information is provided in tables and does not all need to be reported again in words. Select the most pertinent points only. Suggest you also report the overall proportion of dancers who sustained an injury in the past 12 months at the beginning of line 159. This is a common outcome measure used in the dance injury literature and could be used to contextualise your findings with other studies and/or genres. Please also include the total number of injuries in the results section (96), as you have in the discussion section. The meaningfulness of your ‘absence’ outcome reported on line 169 is unclear. Suggest deleting and only using the injury severity measure. Table 2: include ‘number of injuries’ as the first row

Discussion

Overall the discussion section could be revised for conciseness Suggest to also include a discussion of the proportion of dancers over the past 12 months in your study, and compare to the literature. You found a higher proportion of traumatic injuries in your study compared to much of the existing dance literature (which has been focussed on ballet). You may wish to include this as a discussion point, and highlight the importance of studying other genres of dance, as findings from one style are not necessarily applicable to ‘dance’ in general. 96% of dancers needing to pay for their own treatment is an interesting finding. Suggest you include some brief discussion of this into your discussion section Suggest including a limitation of sample size, there were only 96 injuries, and therefore some of your non-significant findings (for example a difference in traumatic vs overuse injuries between males and females) may be due to a lack of power to detect a difference. As previously mentioned, dancing less than 1000 hours per year does not preclude presentation of the results as per 1000 hours. As you have collected dance hours data, I do suggest reporting your incidence findings in this format instead. Especially since the current presentation of incidence in table 2 is so small. Your conclusion statement focuses on current training practices, however the original purpose of your study was to “examine retrospectively the type of injury, body region, incidence, severity, injury sustained according to the type of activity and perceived cause of injuries between gender and age-class in DanceSport in the last12 months”. Suggest you revise this section for better alignment.

All the best with you research

Author Response

Response to Reviewer 2 Comments

Dear reviewer,

Thank you very much for your feedback and suggestions. We try to improve our paper as much as we could and hopefully, our changes satisfied your expectation.

Introduction:

There is contention in the field regarding the classification of dance as only a performing art and not a competitive sport. Therefore I suggest deleting lines 29-30, and focusing only on explaining the role competitions play in DanceSport, rather than comparing this with ballet and modern dance. Overall the introduction is lengthy and at times the relevance to your specific research questions are unclear. Suggest to review and tighten this whole section. References need to be updated, as the most recent reference is from 2014. Some Dance Sport injury studies have been published since this time that are of relevance, including: Pellicciari et al 2016 “Injuries Among Italian DanceSport Athletes: A Questionnaire Survey”, and Riding McCabe et al 2014 “Fit to dance survey: a comparison with dancesport injuries” and Riding McCabe et al 2013 “A Bibliographic Review of Medicine and Science Research in DanceSport”

We improved the introduction part according to all reviewers suggestions. Thank you for suggested literature. We included it in the paper.  

Methods:

Use of reference 26 on line 97: there is likely a more relevant reference to support the questionnaire used in the study (Fit to Dance perhaps?) Dance training volume information is duplicated in both the demographic data and the injury history sections, only needs to be in one. Please provide more information on how ‘hours danced in the last 12 months” was determined (for example was it the average hours spent dancing per week X weeks spent dancing per year?) Lines 122-25: presentation of injuries as a rate per 1000 dance hours provides standardisation of results across injury epidemiology studies. I disagree that it cannot (or should not) be applied to participants who train less than 1000 hours per year. Suggest deleting this section Severity of injury: please provide more details as to how many days off dance would be required for each severity ‘group’? Or were these data collected and analysed as a continuous variable? If so I suggest rewording this sentence, as data were not ‘grouped’.

According to your suggestion injuries were reported per 1000h of dancing. Injury definition, injury incidence and severity, site of injury etc was modified from studies by Brooks et al, Fuller et al, and Pluim et al. Similar methodological approach was also used in previous studies (Allen et al, Angioi et al).

Results:

Wording of the results section could be refined down quite a bit, much of this information is provided in tables and does not all need to be reported again in words. Select the most pertinent points only. Suggest you also report the overall proportion of dancers who sustained an injury in the past 12 months at the beginning of line 159. This is a common outcome measure used in the dance injury literature and could be used to contextualise your findings with other studies and/or genres. Please also include the total number of injuries in the results section (96), as you have in the discussion section. The meaningfulness of your ‘absence’ outcome reported on line 169 is unclear. Suggest deleting and only using the injury severity measure. Table 2: include ‘number of injuries’ as the first row

All results about injury are now reported per 1000h of dancing. Overall proportion of dancers who sustained an injury in the past 12 months is added. Absence was reported since the previous studies also included this in their research.

Discussion

Overall the discussion section could be revised for conciseness Suggest to also include a discussion of the proportion of dancers over the past 12 months in your study, and compare to the literature. You found a higher proportion of traumatic injuries in your study compared to much of the existing dance literature (which has been focussed on ballet). You may wish to include this as a discussion point, and highlight the importance of studying other genres of dance, as findings from one style are not necessarily applicable to ‘dance’ in general. 96% of dancers needing to pay for their own treatment is an interesting finding. Suggest you include some brief discussion of this into your discussion section Suggest including a limitation of sample size, there were only 96 injuries, and therefore some of your non-significant findings (for example a difference in traumatic vs overuse injuries between males and females) may be due to a lack of power to detect a difference. As previously mentioned, dancing less than 1000 hours per year does not preclude presentation of the results as per 1000 hours. As you have collected dance hours data, I do suggest reporting your incidence findings in this format instead. Especially since the current presentation of incidence in table 2 is so small. Your conclusion statement focuses on current training practices, however the original purpose of your study was to “examine retrospectively the type of injury, body region, incidence, severity, injury sustained according to the type of activity and perceived cause of injuries between gender and age-class in DanceSport in the last12 months”. Suggest you revise this section for better alignment.

The discussion, limitation and conclusion were rewritten according to your suggestions.

Thank you very much for all your suggestions that help us to improve our paper. We hope that we satisfied your expectations.

Kind Regards

Reviewer 3 Report

see attached file

Author Response

Response to Reviewer 3 Comments

Dear reviewer,

Thank you very much for your feedback and suggestions. We try to improve our paper as much as we could and hopefully, our changes satisfied your expectation.

Introduction:

We improved the introduction part according to all reviewers suggestions.

Materials and Methods

Line 90: there are only 97 participants in 5 groups. This makes a serious statistic impossible, because the group sizes become too small.

Thank you very much for this suggestion.

Although we do agree in principle with your comment in regards to making worthwhile statistical inferences from smaller sample sizes we feel, that due to the nature of sportdance and specifically to these pre-determined age classifications within this sport; it will be best if we leave the age classifications as it stands. This is because these age-classes are based on the age dance classification and it will make more sense to the dance community and the results will have more applicability. Furthermore, each age-class is engaged in different training exercises and routines which might also be associated with different types of injuries, different body sites and therefore different incidence of injuries.

Which questionnaire has been used?

We used the questionnaire form Fuller et al, and adapted it a little bit to this study’s objectives. Injury definition, injury incidence and severity, site of injury etc was modified from studies by Brooks et al, Fuller et al, and Pluim et al. Similar methodological approach was also used in previous studies (Allen et al, Angioi et al) Distinction between chronic and acute injuries was also made according to the definition use by Brooks et al and Fuller et al.

Fuller CW, Ekstrand J, Junge A, et al. Consensus statement on injury definitions and data collection procedures for studies of injuries in rugby union. Br J Sports Med. 2006;40:193-201. Brooks JH, Fuller CW, Kemp SP, Reddin DB. Epidemiology of injuries in English profes­sional rugby union: part 2 training injuries. Br J Sports Med. 2005;39:767-775. Pluim BM, Fuller CW, Batt ME, et al. Consen­sus statement on epidemiological studies of medical conditions in tennis, April 2009. Br J Sports Med. 2009;43:893-897. Allen N, Nevill A, Brooks J, Koutedakis Y, Wyon M. Ballet injuries: Injury incidence and severity over 1 year. J Orthop Sports Phys Ther. 2012;42:781-790 Angioi M, Metsios S, Koutedakis Y, Twitchett E, Wyon M. Physical fitness and severity of injuries in contemporary dance. Med Probl Perform Art. 2009;24:26-29.

The period of 12 months retrospectively is too long. This is not even discussed as a limitation. How is it excluded that not only those who are injured participate, but all?

We are aware that the questionnaires were completed by participant recall and that this is a limitation of our study, but most published retrospective studies have examined the 12 months period. We added this in our limitation part. Although we are not entirely clear with the meaning of the second part of this comment, we will try to answer it by describing concisely the methods of this study. Our inclusion criteria did not include only dancers that had been injured. This study’s questionnaires were provided to all the dancers we had access to during these International competitions, therefore everybody was invited to participate into our study independent of if they were injured or un-injured currently or within the past 12 months.

The discussion contains numerous parts from the results. These must be removed. In addition, it is unclear why the results are discussed in relation to other dance directions - this was not the question and it is also not stated why this procedure is chosen. In the introduction only the differences of the dance directions are pointed out.

The discussion must therefore be completely revised. Limitations: Too short, incomplete. Conclusions: Cannot be derived from the results in this way. Has to be rewritten.

The discussion, limitation and conclusion were rewritten.

Thank you very much for all your suggestions that help us to improve our paper. We hope that we satisfied your expectations.

Kind Regards